# Chia, Quinoa, and Their Coproducts as Potential Antioxidants for the Meat Industry

**DOI:** 10.3390/plants9101359

**Published:** 2020-10-14

**Authors:** Juana Fernández-López, Manuel Viuda-Martos, María Estrella Sayas-Barberá, Casilda Navarro-Rodríguez de Vera, Raquel Lucas-González, Alba Roldán-Verdú, Carmen Botella-Martínez, Jose Angel Pérez-Alvarez

**Affiliations:** IPOA Research Group, Agro-Food Technology Department, Higher Polytechnic School of Orihuela, Miguel Hernández University, Orihuela, 03312-Alicante, Spain; j.fernandez@umh.es (J.F.-L.); mviuda@umh.es (M.V.-M.); estrella.sayas@umh.es (M.E.S.-B.); casilda.navarro@umh.es (C.N.-R.d.V.); raquel.lucasg@umh.es (R.L.-G.); alba.roldan1880@gmail.com (A.R.-V.); c.botella@umh.es (C.B.-M.)

**Keywords:** chia, quinoa, antioxidants, meat products, lipid oxidation

## Abstract

Chia and quinoa have gained popularity among consumers worldwide due to the wide variety of nutrients but also to the bioactive compounds that they contain. Lately, their processing has generated different coproducts (non-commercial grains, flour, partially deoiled flour, rich-fiber fraction, and oil, among others), which could be reincorporated to the food chain with important technological properties, antioxidant activity included. Both sets of ingredients have been revealed a great technological potential for meat product development and innovation, taking into account that oxidation is one of the main reactions responsible for their deterioration and shelf life reduction. This review focuses on the antioxidant compounds of chia and quinoa coproducts and on the strategies used to add them to meat products highlighting their effect on the lipid oxidation control. Apart from the different ways in which quinoa and chia can be incorporated into meat products and their antioxidant properties, innovative approaches for increasing this antioxidant effect and counteracting any negative alterations they may cause will be discussed.

## 1. Introduction

Chia (*Salvia hispanica L.*) and quinoa (*Chenopodium quinoa* Willd) seeds have been consumed in their respective country of origin for centuries (Andean countries for quinoa and Mexico and Guatemala for chia); however, currently they have gained a renewed relevance in developed countries due to their excellent nutritional value and also to the large variety of bioactive compounds that they contain [1,2,3,4]. In addition, both seeds have demonstrated interesting technological properties in food processing that make them not only suitable for their direct consumption but also for their application as ingredients in different foods [5,6,7,8,9]. Due to the boom in the consumption and commercialization of chia and quinoa related food products, which includes not only quinoa and chia seeds but also different types of foods in which both can be applied, in the form of flours (obtained by a dry or wet-milling process), bran, enriched fractions (fiber, protein, and starch), oil, mucilage, partially deoiled flour, etc., depending on the aim of the food industry, a wide range of coproducts are been generated [7,9,10,11] (Figure 1). Chia and quinoa (seeds or flour) can be added or mixed into bread, biscuits, pasta, snacks, cakes, emulsified meat products, etc., as supplements or as substitutes for eggs and fat. Quinoa flour has also been used as binder or extender in meat products. Fiber, protein, and starch-rich fractions from quinoa can be used to increase the content of these nutrients in several foods. Chia mucilage may be used as a foam stabilizer, emulsifier, or binder in the food industry and also as a functional coating with improved functional properties. Chia oil can be used to replace animal fat in foods with an increase in their nutritional value. Partially deoiled chia flour has been used as a source of dietary fiber in meat products. All of them are examples about the application of some of these coproducts in the food industry [5,9,11,12,13,14].

Nutritionally, quinoa stands out for its fiber content (5–11%), starch (52–74% being rich in amylopectin), proteins (13–17%) with the presence of essential amino acids (methionine and lysine but also a high concentration of tryptophan, usually the second limiting amino acid in cereals), and fatty acids (2–9% oil with 90% unsaturated fatty acids of which 50–56% correspond to linoleic acid omega-6, 22–25% to oleic acid omega-9, and 5–7% to linolenic acid omega-3), absence of gluten, and adequate levels of key micronutrients, including minerals (calcium, magnesium, iron, potassium, phosphorus, manganese, zinc, copper, and sodium) and vitamins (vitamin B complex, E, and C) [8,15,16,17]. Quinoa also contains bioactive constituents, such as carotenoids, tocopherols, polyphenols, and betalains, which are responsible for protecting the plant against adverse climatic conditions and are highly correlated with its antioxidant activity [4,18,19]. Regarding chia, it stands out also for its fiber content (18–30% being soluble dietary fiber 7–15%), proteins of high biological value (15–25%), and it is considered a natural source of omega-3 (α-linolenic acid up to 68% of fatty acid content) and omega-6 fatty acids (linoleic acid 19%) [20,21,22]. It also contains bioactive compounds such as tocopherols, polyphenols, vitamins (mainly vitamin B1, B2, and niacin), and carotenoids [23,24,25].

Most of the bioactive compounds in quinoa and chia seeds are related to their antioxidant activity. Antioxidants are substances that protect cells against oxidative damage caused by excess reactive oxygen species. Oxidative stress, which releases free oxygen radicals in the body, has been implicated in several disorders, including chronic-degenerative diseases such as cardiovascular diseases, cataracts, diabetes, Alzheimer’s disease, cancers, and rheumatism [26,27,28,29]. It is well known that bioactive compounds in vegetal foods can act as antioxidants, protecting the cells. On the other hand, the addition of chia or/and quinoa in foods enables that their antioxidant compounds could prevent food deterioration caused by lipid oxidation. Lipid oxidation in foods (oils, fats, and other fat-containing foods) is responsible for the development of primary and secondary oxidation products, reduction in nutritional quality, as well as changes in flavor [30,31], which can precipitate health hazards and economic losses in terms of inferior product quality [32,33].

Specifically, in meat and meat products, lipid oxidation is the major cause of quality deterioration, affecting either the stored triglycerides or the tissue phospholipids. In addition, ferric heme pigments have been implicated as the major pro-oxidants in meat lipid oxidation, being both reactions, pigment, and lipid oxidation, interrelated [34,35]. Nonheme iron may also function as prooxidant in meat [36]. Additionally, even some processing operations during meat products manufacturing (grounding, sodium chloride addition, and cooking, among others) contribute to accelerating the oxidation of the triglycerides [37,38]. For these reasons, antioxidants (synthetics) have been usually added to fresh and processed meat and meat products to prevent lipid oxidation, retard the development of off-flavors, and improve color stability by scavenging chain-carrying peroxyl radicals or suppressing the formation of free radicals [39]. The most widely used synthetic antioxidants in the meat industry, butylated hydroxyanisole (BHA) and butylated hydroxytoluene (BHT), are quite volatile and decompose easily at high temperatures [40]. There are serious concerns about the safety and toxicity of BHA, BHT and *tert*-butylhydroxyquinone (TBHQ) related to their metabolism and possible absorption and accumulation in body organs and tissues [41]. In response to the claims that synthetic antioxidants have the potential to cause toxicological effects and consumers increased interest in purchasing natural products, the meat industry has been seeking sources of natural antioxidants [33]. Most of these natural antioxidants came from fruits, spices or other plant-based ingredients (which may be found in any plant part, such as grains, fruits, nuts, seeds, leaves, roots, arils, and barks) and they owe their antioxidant activity mainly to their bioactive compounds (polyphenols, anthocyanins, tocopherols, and carotenoids, among others) [42,43].

Considering that, in this review, we summarize and discuss recent advances about the composition and identification of bioactive compounds with antioxidant properties in chia and quinoa edible grains and their coproducts. It is also aimed at exploring the effect of the addition of chia, quinoa or their coproducts on the development of the lipid oxidation in chia and quinoa related meat products due to the antioxidant properties attributed to these bioactive compounds.

## 2. Antioxidants Compounds in Chia, Quinoa, and Their Coproducts

There are a lot of works about the bioactive compounds in chia and quinoa seeds that confirm that they contain a rich pool of natural antioxidants such as tocopherols, phytosterols, carotenoids, and phenolic compounds [4,8,18,44,45,46,47,48,49], which could protect consumers against many diseases and also promote beneficial effects on human health [50].

In both cases (chia and quinoa), it has been reported that climatic conditions and geographical location of growing have a great influence on the chemical composition of the seeds [23,47,51,52,53,54,55,56]. In addition to the differences in the composition of the several bioactive compounds, it is necessary to add another variation factor such as the extraction procedure used (solvent, procedure, and extraction time) since it is a critical process for some matrices, particularly when there may be insoluble components with antioxidant capacity, which lead to underestimated these values in some cases [57,58,59]. Recently, important differences between authors have also been detected depending on the application or not of a hydrolysis treatment, which can be used for different purposes depending on when it is applied; if it is used during the extraction process, it can help to elucidate the amount of the phenolic compounds bounded to other structures in the cell wall, which are not easily extracted, but if it is used to the extract prepared it can simplify the sample and make easier the identification step [23,60]. In addition, depending on the coproduct, also important differences in the content on bioactive compounds could be expected. All these variability factors make it difficult, in some cases, to compare data reported by different authors, however, similar trends can be observed in most of them (Table 1).

### 2.1. Chia and Its Coproducts

In reference to chia seeds, most of the data already reported in the literature show total phenolic content values ranging between 0.5 and 3.9 mg gallic acid equivalents (GAE)/g chia. Da Silva et al. [56] reported a TPC of 0.98 mg GAE/g in chia seeds from Brazil, which is in accordance with the amount reported by other authors for chia seeds from Chile (0.94 mg GAE/g, [47]; 1.16 mg GAE/g, [60]), Mexico (0.90 mg GAE/g; [23]), or Bolivia (1.16 mg GAE/g; [60]). Higher TPC values (3.9–4.1 mg GAE/g chia) have been reported by Pellegrini et al. [3] and Fernández-López et al. [22]. Alcântara et al. [55] reported a wide variation range (10.1–60.9 mg GAE/g chia) in chia seeds from Brazil depending on the solvent used. Controversial results have been reporting regarding the effect of the hydrolysis process; Reyes-Caudillo et al. [23] reported that the acidic hydrolysis did not increase the TPC compared to crude extract of chia seeds, whereas Oliveira-Alves et al. [60] reported a significant increase in hydrolyzed extracts. It is important to highlight that after the extraction of chia oil, only 5% of TPC was found in chia oil and the rest (95%) remain in the deoiled chia flour [3,22,60]. TPC in chia oil also depend on the oil extraction procedure: Itxaina et al. [78] reported that TPC was significantly higher when the oil was obtained by pressure than by the solvent system. Xuan et al. [79] compared the TPC of 14 edible oils (chia, flax, sunflower, canola, grape, avocado, and sesame, among others) reporting for chia oil one of the lowest values (4.86 mg GAE/g oil; similar to sunflower oil) in comparison with the TPC found in flax (39.2 mg GAE/g oil) or avocado (11.31 mg GAE/g oil) oils. On the other hand, the partially deoiled flour obtained after the extraction of oil from chia seeds has a TPC similar than reported for chia seeds [22].

About the composition of the phenolic compounds in chia seeds, in all cases, different phenolic acids, flavonoids, and tannins have been identified. Figure 2 shows the main antioxidant compounds identified in chia seeds. Rosmarinic acid has been reported as the major compound detected and quantified in chia seeds [3,22,25,55,60] representing approximately 75–90% of the phenolic acids. Ferulic, caffeic, salicylic, and the protocatechinic acids have also been identified [3,23,25,46,55]. The carnosol, a phenolic diterpene, has been identified by Oliviera-Alves et al. [60]. Ding et al. [21] reported that flavonoid content occupied 80.8% in the polyphenols of chia, being rutin and hesperidin the major components. The flavonols group has been also identified in most of the works, being myricetin, quercetin, and kaempferol (mainly glycosides but also in aglycon forms) the predominant ones [18]. Myricetin was also the second main compound identified by Alcântara et al. [55] representing 88% of the flavonoids. Pellegrini et al. [3] reported quercetin as the main flavonoid and also detected myricetin and rutin. The isoflavones group has also been mostly identified. Martínez-Cruz and Paredes-López [25] identified daidzin, glycitin, genistin, glycetein, and genistein and proposed the chia seeds as a novel isoflavone source in the human diet. Pellegrini et al. [3] and Fernández-López et al. [22] identified daizdin as the second compound most abundant (followed by quercetin), and genistin, and genistein were also identified. On the contrary, Reyes-Caudillo et al. [23] reported that the anthocyanins group was not detected in Mexican seeds. Ixtaina et al. [45] reported that the major phenolic compounds in chia oil were chlorogenic and caffeic acids, followed by myricetin, quercetin, and kaemferol, being all of them also detected in chia seeds. It is interesting to note that most of the phenolic compounds found in chia are not present in other oilseeds [47,79]. Fernández-López et al. identified, in partially deoiled chia flour, the same 11 compounds (phenolic acids, flavonols, and isoflavones) than in the corresponding chia seeds, although in higher concentrations, which confirms the fact that most of the phenolic compounds remain in this coproduct after the oil extraction. Similar results have been reported for this chia coproduct by Capitani et al. [46]: in this case, the authors did not find differences in the total polyphenolic component concentration between the partially deoiled flours obtained for solvent or pressing oil extraction.

Tocopherols have also found in chia seeds although the concentration and the type (isoform) identified vary depending on growth location and conditions. Da Silva et al. [56] reported in chia seeds from Brazil, an average content of total tocopherols (α-, β-, γ-, and δ-) of about 8205.6 µg/100 g, being γ-tocopherol the predominant isoform (90% approximately). Capitani et al. [46] reported a similar tocopherol concentration in chia grown in Argentina and also in this case the component present in larger amounts was γ-tocopherol. Controversial results were observed in other studies that used chia grown in Argentina and Guatemala [45,46] since the authors did not detect the presence of β-tocopherol. Itxaina et al. [45] reported approximately 300 mg of tocopherols/kg in the chia oil obtained by solvent and lower amounts (238 mg/kg) when it was obtained by pressing, which is in accordance with that reported by Guiotto et al. [78]. Capitani et al. [46] also detected tocopherols in partially deoiled flours obtained by pressing and solvent extraction being γ-tocopherol the main component (approximately 95%) in both cases.

Carotenoids are minor phytochemicals detected in chia seeds and only a few studies have been developed to identify them, so it could be said that chia carotenoid composition is still unknown. Da Silva et al. [56] reported a total amount of carotenoids of 57 µg/100 g in chia seeds from Brazil, being the zeaxanthin identified as the main compound. Dabrowski et al. [80] reported that ca. 2/3 constitutes lutein, with and approximately 30% share of β-carotene and small amounts of 9-cis-β-carotene.

### 2.2. Quinoa and Its Coproducts

Quinoa grains also are considered a good source of various bioactive compounds with antioxidant properties being the phenolic compounds the most important and studied although recently, several carotenoids and betalains, mainly present in colored quinoas, have also been highlighted due to their antioxidant properties. All these compounds contribute to the antioxidant effect of quinoa grains. Figure 3 shows some of the principal antioxidant compounds identified in quinoa seeds. Polyphenols are natural organic chemicals with large multiples of phenol structural units. The TPC reported in quinoa grains ranged between 0.2 and 10.6 mg GAE/g quinoa with significant differences depending (in addition to the other variation factors previously discussed) on their color: darker quinoa seeds (such as red and black quinoas) showed higher TPC than white varieties [4,18,48,49,67]. Not only the TPC was suggested to differ according to the color of quinoa grains, but also the phenolic profiles and their antioxidant activity. Recently, several authors have reported that the malting process increases the amount of phenolic compounds (up to 49% comparing to the initial content before malting) in quinoa flour [71]. These authors also reported that this increase was higher in flour from colored quinoas (red and black) than from the white ones. Ballester-Sánchez et al. [10] determined the TPC (both polyphenol fractions, extractable and hydrolysable) in red quinoa seeds and in the fiber-rich fractions obtained by the dry and wet-milling process, reporting values for the extractable fraction (the most frequently reported by authors) of 4.9 mg GAE/g for the whole seeds and 3.8 mg GAE/g and 6.6 mg GAE/g for the fractions from wet and dry-milling, respectively. As can be seen, the wet-milling fiber showed lower values in the extractable polyphenol fraction than the whole seeds, whereas they were the same in the case of the fiber fraction from dry-milling. These results were expected taking into account that the extractable polyphenol fraction consists of polyphenols water soluble and can therefore be leached in the steeping water used during the wet-milling process. These values were significantly increased when the hydrolysable fraction was assessed, reporting values of 34, 77, and 83 mg GAE/g for the whole seeds, dry-milling, and wet-milling fractions, respectively. It is important to note that in both fiber-rich fractions the hydrolysable polyphenol fraction was clearly enriched in comparison with the quinoa seeds (around 2.5 fold). This result also was expected given that this fraction (hydrolysable polyphenols) represents the polyphenols bound to cell wall macromolecules and to dietary fiber. The most abundant polyphenols in quinoa are phenolic acids and flavonoids [2,18,65]. Eleven kinds of phenolic acids have been detected in quinoa samples (protocatechuic, *p*-hydroxybenzoic, vanillic, syringic, *P*-coumaric, ferulic, sinapic and isoferulic, cholorogenic, rosmarinic, and caffeic acids) with their total amount ranging from 56 to 95 μg/g [4,18,49,65,81]. Phenolic acids also exist in the form of free and bound and it has reported significant differences in the content of bound phenolics between different quinoa varieties, while their content of free phenolics showed relatively small differences [26,82,83]. Ferulic acid and its derivatives were the predominant phenolics in the bound form to be present in quinoa seeds [4,81,84]. It is relevant because bound phenolics of quinoa have shown a higher ability to scavenge the 2,2-diphenyl-1-picrilhidrazilo (DPPH) and the 2,2-azino-bis(3-ethylbenzothiazoline-6-sulfonic acid) (ABTS) free radicals than the free phenolics [49,82].

The total flavonoid content in quinoas ranged between 1.77 and 19.29 mg rutin equivalents/g quinoa and also in this case, red and black quinoas showed the highest values [4,49,85]. Quercetin, isoquercetin, rutin, hesperidin, neohesperidin, catechin, and epicatechin, among others, are flavonoids with antioxidant activity detected in quinoa samples, being quercetin who shows the strongest antioxidant among the flavonoids [18,83,86]. Stickit et al. [81] detected four flavonoid compounds (aesculin, phorizin, coniferyl aldehyde, and eriodictyol) and one stilbene (pterostilbene) for the first time in quinoa seeds grown in Denmark, but at very low concentration (0.08–0.53 mg/kg dw).

Tocopherols belong to a class of phenolic antioxidants that can inhibit lipid autoxidation by scavenging free radicals and by reacting with singlet oxygen. The total tocopherol content reported for quinoa ranged from 37.49 to 59.82 µg/g [8,18,87]. All four tocopherol isoforms (α, β, γ, and δ) have been detected in quinoa seeds being γ tocopherol the predominant isoform in the three varieties (black grains showed the highest concentrations, followed by the red varieties and lastly the white varieties), followed by α-tocopherol, β-tocopherol and δ-tocopherol was the least [18]. Although the most famous antioxidant power has been attributed to the isoform α-, recently, it has been reported that γ-tocopherol is a strong anti-inflammatory agent with equal or even stronger antioxidant properties than the isoform α.

Carotenoids can be not only provitamin A but also strong antioxidants with various health-promoting properties. The total carotenoid content in quinoa seeds range between 11.87 and 17.61 μg/g, showing black quinoa as the highest values [18]. Lutein and zeaxanthin are the two main carotenoids, with lutein being the dominant carotenoid in quinoa seeds. Tang et al. [88] reported that most quinoa seeds also contain β-carotene.

Betalains are nitrogenous compounds soluble in aqueous media according to their chemical structure, these pigments can be subdivided as red-violet betacyanins or as yellow-orange betaxanthins. Their presence has been reported in colored quinoa grains (yellow and red-violet) [18,19,26]. Betalains show high antioxidant and free radical scavenging activities supported by the high antiradical capacity of the pigments’ structural unit, betalamic acid [83,89,90]. Escribano et al. [19] evaluated the antioxidant activities of quinoa grains of multiple colors using the FRAP and ABTS assays, reporting that FRAP results indicated a very high antioxidant capacity for the pigmented quinoa samples in comparison with the white and black ones. In addition, they reported that the highest activity was observed for the red-violet varieties, which contain both betacyanins and betaxanthins.

## 3. Antioxidant Activity in Chia, Quinoa, and Their Coproducts

Data variation in the antioxidant capacity of chia and quinoa is to be expected, as many factors such as genetics, agrotechnical processes, and environmental conditions can influence the presence of phenolic compounds [91] as has been mentioned above (point 2). In addition, a comparison of results from different studies can be difficult due to variability in the experimental conditions amongst the methods used for their evaluation [92]. Several antioxidant capacity assays have been applied to evaluate the chemical mechanism involved in the antioxidant action. Depending upon the reactions involved, these assays can roughly be classified into two types: assays based on hydrogen atom transfer (oxygen radical absorbance capacity (ORAC) assay and total radical trapping antioxidant parameter (TRAP), and assays based on electron transfer (Trolox equivalence antioxidant capacity (TEAC) assay or ABTS assay, ferric ion reducing antioxidant power (FRAP) and the DPPH free radical method). In addition, other methods are used to evaluate their chelating activity on specific pro-oxidants, such as the method to measure the ferrous ion chelating activity (FIC) [92]. In other words, it could be said that the FRAP assay measures the ability of quinoa and chia seeds to reduce ferric ion, DPPH, and ABTs assays measure the radical quencher ability and the FIC assay the ability to quench iron.

### 3.1. Chia and Its Coproducts

Considering the molecular structure of the major compounds identified in chia seeds, a high antioxidant activity would be expected: rosmarinic acid and myricetin, the first consists of two aromatic rings and five hydroxyl groups, and the second consists of three aromatic rings with six hydroxyl groups. Thus, these compounds provide high availability of the hydroxyl groups to react with free radicals or reduce other compounds. Pekkarinen et al. [93] showed that compared with other flavonoids (quercetin, kaempferol, catechin, and rutin), myricetin showed the highest antioxidant capacity. Studies prove that myricetin has more phenolic hydroxyl groups, indicating that its antioxidant capacity increases with the number of these hydroxyl groups. So, the high antioxidant activity of chia seeds can be mainly attributed to their high content of phenolic compounds (and also to tocopherols) and it can be developed through to various mechanisms, among which are the prevention of chain initiation, binding of transition metal ion catalysts, decomposition of peroxides, prevention of continued hydrogen abstraction, and radical scavenging protecting against oxidative damage to DNA, proteins, and lipids [94]. Most of the authors reported a high antioxidant activity in chia seeds, comparable to Trolox^®^, suggesting that the phenols in these extracts have an important activity as oxygen singlet quenchers [3,23,47,55]. The antioxidant activity of chia seeds has been assessed using different methods, being the most usual ORAC (489–517 μmol TE/g), FRAP (18.5–71.8 mg TE/g), and DPPH (5.4–49.4 mg TE/g) [3,22,60,74] (Table 1). Several authors have reported that the antioxidant activity of chia oil is much lower than chia seeds although it was not proportional to the decrease of the TPC, suggesting that other compounds not detected using these analytical conditions could be responsible [60]. The results reported by Xun et al. [77] seem to confirm this, because the DPPH scavenging activity reported for different edible oils was not totally related with the TPC; in the case of chia oil the DPPH value was the lowest, together with avocado oil. In reference to that, Itxaina et al. [45] reported that the high level of polyunsaturated fatty acids (PUFAs) would be the main cause for the low oxidative stability prevailing over the antioxidant effects associated with bioactive components (tocopherols, polyphenols, carotenoids, and phospholipids) present in chia seed oils. Regarding the partially deoiled flour, several authors have found a higher antioxidant activity than in the raw material [3,22]. Vazquez-Obando et al. [95] reported a high antioxidant activity for the chia fiber-rich fraction, (ABTS: 488.8 μmol TE/g) similar to that of sorghum bran with high tannin content and higher than those for some wheat grains and sorghum.

### 3.2. Quinoa and Its Coproducts

Many researchers have attributed the antioxidant activity of quinoa seeds only to their hydrophilic phytochemicals, mainly phenolics, but in some cases also betalains [19,26]; however, the contribution of lipophilic compounds (tocopherols and carotenoids) must not be underestimated [88]. In reference to the antioxidant properties of quinoa grains (black, white, and colored varieties) attributed to their phenolic compounds, they were assessed using different methods, being the most usual DPPH (0.1–9.7 mg TE/g), ABTS (3.88–7.76 mg TE/g), FRAP (0.7–9 mg TE/g), and FIC (0.59–0.97 μg EDTA/g) assays (Table 1). These results showed a good correlation with phenols and flavonoids content [4,59,81]. For all assays, red and black quinoas showed better antioxidant activities than white varieties [4]. In view of these results, it could be indicated greater importance of the antioxidant-antiradical power in comparison with the electron transfer and reducing iron power in quinoa [81]. The antioxidant activities of the lipophilic extracts in quinoa seeds were evaluated by DPPH (6.26 μmol TE/g), FRAP (6.76 μmol AAE/g), and ORAC (6.69 μmol TE/g) activities and these results also showed good and significant correlations with carotenoids and tocopherols content [88].

Ballester-Sánchez et al. [10] reported higher antioxidant capacity (assessed by DPPH and FRAP analysis) in the fiber-rich fractions (both, obtained by wet and dry-milling process) than in the whole seeds (red quinoa; Table 1). Additionally, in this case the antioxidant activity was much higher in the hydrolysable polyphenol fraction than in the extractable (DPPH: 10.5 fold in whole seeds, between 54 and 40 fold in fiber-rich fractions; FRAP: 6 fold in whole seeds, between 42 and 24 fold in fiber-rich fractions).

## 4. Oxidation Stability of Meat Products Containing Chia, Quinoa, or Their Coproducts

Focusing on reports from the recent 5 years, quinoa and chia-related meat products seem to have aroused great interest in the meat sector not only for the improvement in the nutritional quality of reformulated meat products but also for their beneficial effect on the technological properties and shelf life [9,24]. One of the most relevant effects that has been addressed in most of these works has been the effect on lipid oxidation in the meat product, not only on the fresh product but also during its storage, attributing these antioxidant effects to the bioactive compounds with antioxidant properties reported for quinoa and chia seeds [21,96,97,98,99]. This oxidation stability on the meat product has been commonly assessed by the reduction in the thiobarbituric acid reactive substances (TBARs values) that has been revealed as the most suitable method for monitoring lipid oxidation in meat and meat products [100]. Table 2 shows chia and quinoa-based meat products and their effect on the lipid oxidation.

### 4.1. Chia Related Meat Products

Chia has been applied in several meat products (burgers, nuggets, fresh pork sausages, frankfurter-like sausages, and restructured ham-like products, among others) not only as whole chia seeds but also as flour or partially deoiled flour (after oil extraction) [9,21,75,96,101,103,114,116,119,120]. This addition was taken off with two main objectives: to partially replace animal fat or to improve their nutritional and techno-functional quality. The addition of 0.5 and 1% chia seeds in a restructured ham-like product decreased lipid and protein oxidation during 4 weeks of refrigerator storage, attributing this antioxidant effect to the polyphenols in chia [21]. This effect was dependent on the percentage of chia addition: the higher the amount of chia added the lower TBARs values. The addition of chia seeds (up to 8%) to partially replace animal fat (pork backfat) in chicken hamburgers [103] was effective no only to reduce fat content but also lipid oxidation. Additionally, in this case, the TBARs values of the cooked burgers were dependent on the amount of chia seeds added (2%, 4%, and 8%): higher additions resulted in lower TBARs values. In the same way, Antonini et al. [75] reported that the addition of chia seeds (2.5% and 5%) to beef burgers provided a significant reduction in malonaldehyde levels compared to control cooked burgers, though not in a dose-dependent manner. These authors evaluated the polyphenol patterns and antioxidant capacities in these cooked burgers reporting a good correspondence between then, especially for the ORAC and ABTS assays. Frankfurter-type sausages with 3% chia seeds added showed greater resistance to fat oxidation than control [120]. These authors also reported this antioxidant effect (at the same level as reported for chia seeds) on frankfurter-type sausages when partially deoiled chia flour was added. As has been previously reported, most of the phenolic compounds identified in chia seeds remain in the partially deoiled chia flour after the oil extraction, being responsible for the antioxidant properties. Low-fat frankfurter with chia flour added (10%) showed an increase in the oxidative stability during chilling storage, which was attributed to the antioxidant compounds in chia [101]. It is important to highlight that most of these meat products in which the antioxidant effect of chia seeds has been reported are cooked meat products, which means that chia seeds retained its antioxidant effect during cooking [103]. Scapin et al. [121] studied the effect of chia seed extract at concentrations of 0%, 1%, 1.5%, and 2% as an antioxidant in fresh pork sausage. They reported TBARs values, after 28 days of storage, of 1.12 mg MDA/kg for the treatment with 2% chia extract and 1.64% mg MDA/kg for the control treatment, which is showing the effect of inhibiting lipid oxidation, suggesting their use as natural antioxidant in meat products.

Chia oil has attracted more and more attention for meat researchers to be used as partial replacement of animal fat mainly due to its high content in alpha-linolenic acid (ALA), relatively low levels of saturated fatty acids and does not contain any of the antinutritional compounds or vitamin B6 antagonist factors present in other sources of ALA [122]. The principal problems to the direct incorporation of this oil into meat products are the technological difficulty for their integration in the meat batter, the instability to oxidation (due to the high of unsaturation of fatty acids), and their negative effects on sensorial attributes. To prevent this, several strategies have been developed like microencapsulation with rosemary antioxidants (burgers [104]), conventional oil-in-water emulsions (cooked lamb sausages [107] and frankfurter [102]), double emulsions (meat systems [123]), emulsion gels (burgers [105]; bolognas [117]; and fresh sausages [112]) or hydrogel emulsions (low-fat burgers [106] and lamb sausages [113]). These methodologies allow the chia oil able to be stabilized or immobilized in the protein matrix, reducing the chances of bulk oil physically separating from the structure of the meat product, and so remaining stable during processing and storage. In this sense, Cofrades et al. [123] reported that the presence of chia oil, with the attendant increase in the level of unsaturation of the lipids, promotes lipid oxidation in cooked meat batters, but it also appears to supply some compounds that can help to protect against oxidation (DPPH scavenging activity). However, chia oil is more susceptible to oxidation when it is incorporated in the double emulsion than when added in the liquid form, possibly due to the degree of interaction with meat proteins.

### 4.2. Quinoa Related Meat Products

Quinoa has been mainly applied in meat products (burgers, meatballs, nuggets, bolognas, pâté, dry-cured sausages, etc.) as the techno-functional ingredient (due to its high protein and carbohydrate content) or as a fat replacer (due to its behavior as a fat-like raw material). In the first case (techno-functional ingredient) it means that quinoa can act like extenders (non-meat compounds with considerable protein content), fillers (plant substances with high carbohydrate content), and binders (substances with high-protein content able to bind both water and fat) [97,98,99,100,108,109,110,111,114,115,118,119]. In the second case (fat replacer) quinoa is used to decrease or totally replace the animal fat content in meat products. However, although these have been the main purpose, several authors have reported that their addition helped to control the development of lipid oxidation in the meat product. Özer and Seçen [109] added quinoa flour (up to 10%; in substitution of breadcrumbs) to beef burgers with the corresponding improvement in burger quality and cooking properties. Additionally, they also found that quinoa flour inhibited lipid oxidation during frozen storage. The antioxidant effect of quinoa flour was identified in raw beef burgers at all usage rates (3%, 5%, 7%, and 10%) during −18 °C storage for 90 days. These antioxidant properties of quinoa flour were also observed when these burgers were cooked, attributing this antioxidant effect to the phenolic and flavonoid content in quinoa as a source of free radical scavenging agents [124]. Fernández-López et al. [98,99] investigated the application of black quinoa seeds (as whole seeds or as their fiber-rich fraction obtained as a coproduct from the quinoa wet-milling process) in bologna-type sausages (up to 3%; in substitution of potato starch). These authors reported that it was a feasible strategy for reformulating cooked sausages and maybe a good choice not only for enhancing the nutritional composition of the bolognas but also for their effect on the technological properties, such as to enhance emulsion stability and to decrease lipid oxidation. In this work, bolognas with quinoa added showed lower TBARs values than control, showing bolognas with quinoa added as whole seeds the lowest TBARs value. It must be taken into account that the fiber-rich fraction is obtained as a coproduct from the wet-milling process of quinoa seeds (whose main product is starch), which implies the use of large quantities of water to allow this fractionation processes [125]. In addition, Lin et al. [84] reported that most of the compounds with antioxidant properties in quinoa seeds are located in the inner of the grain, and only a reduced amount has been identified in bran. This antioxidant effect of black quinoa in bolognas was also confirmed during their refrigerated storage, showing sausages with quinoa products added lower TBAR values than the control for all the days studied, and also, in this case, sausages with quinoa seeds showed the lowest TBAR values throughout the entire storage period (21 days) [99].

Pellegrini et al. [118] used white, red, and black quinoa pastes as the “fat-mimetic” ingredient to reduce the fat content (up to 10%) in the pâté. This reformulation not only increased the healthiness of the product (lower fat and higher fiber content) but also showed beneficial effects on their stability (higher emulsion stability and lower lipid oxidation) without negatively affecting microbial and sensorial quality. All pâté samples with 10% quinoa added (white, black, or red quinoa) showed lower TBAR values than the control. However, when they were added at 5%, only samples with black quinoa achieved significantly lower TBAR values than the control. In this case, the higher content in antioxidant compounds reported for black quinoa (in relation to white and red varieties) could explain this behavior. Several authors have reported differences in the content of antioxidant compounds (polyphenols, tocopherols, betalains, carotenoids, etc.) and their antioxidant activity depending on the quinoa color (white, red, black, violet, yellow, etc.) [4,19], which has been previously highlighted and discussed (points 2.2 and 3.2). Fernández-Diez et al. [119] is the only reference found about the use of quinoa (boiled grains) in dry-cured sausages. In this case, the authors estimated the effect of quinoa addition on the lipid oxidation by means of the determination of the volatile compounds. They reported that volatiles formed via lipid oxidation such as straight medium-chain aldehydes and ketones, which represent the majority of the straight medium-chain alkyl compound group, seemed to decrease with fat reduction and quinoa addition.

## 5. Conclusions

The meat industry is demanding antioxidants from natural sources to replace synthetic ones because of the negative concerns regarding some of the synthetic antioxidants. Chia and quinoa seeds and their coproducts, as the same as other plant materials, provide good alternatives due to their content in bioactive compounds (mainly phenolic compounds), which various kinds of biological activities, antioxidant properties included. The main bioactive compounds identified in quinoa products include phenolic acids (mainly rosmarinic and chlorogenic acids), flavonoids (mainly quercetin and isoquercetin), and nitrogen-containing compounds (mainly betalains: betacyanins, and betaxanthins). These last compounds are water-soluble pigments of hydrophilic nature with promising bioactive potential, identified in colored quinoa grains (violet, red, and yellow), becoming one of the scarce edible sources of betalains. On the other hand, chia products have been reported as a good source of phenolic acids (mainly rosmarinic, ferulic, and caffeic acids) and flavonoids (mainly rutin, myricetin, and quercetin). Isoflavones (mainly daidzin, genistin, and genistein) and tocopherols (γ-tocopherol as the predominant one) have also been identified in chia products. All this richness in antioxidant compounds has allowed their application in meat product formulation not only for their improvement in nutritional and technological quality but also for their antioxidant properties mainly shown in the lipid oxidation control. Among other effects, the factors associated with the way in which these ingredients are incorporated to the meat matrix (flour, oil, emulsions, emulsion gels, etc.), the complexity of this matrix (emulsified meat product, cooked meat product, dry-cured meat product, etc.) or the nature of other ingredients in the formulation (sodium chloride, nitrites, phosphates, etc.) could be a limit to the possibilities of establishing clear relationships among parameters of lipid oxidation and antioxidant capacity of the matrix. In any way, chia and quinoa products open a wide range of opportunities for the meat industry to reformulate processed meat products in view of healthier and cleaner meat products production. In addition to optimizing the formulation to avoid technological problems and to ensure a positive sensorial evaluation of the final meat product, it could be interesting to assess the effect of digestion process on the stability and bioaccessibility of these bioactive compounds and their antioxidant properties using in vitro digestion studies.

## Figures and Tables

**Figure 1 plants-09-01359-f001:**
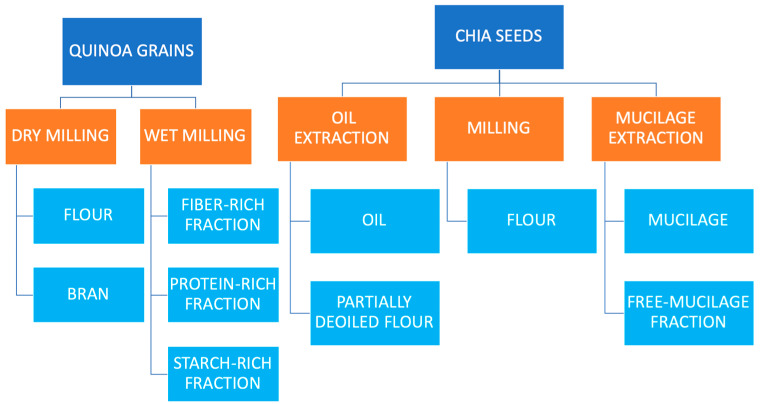
Main coproducts obtained from quinoa and chia processing.

**Figure 2 plants-09-01359-f002:**
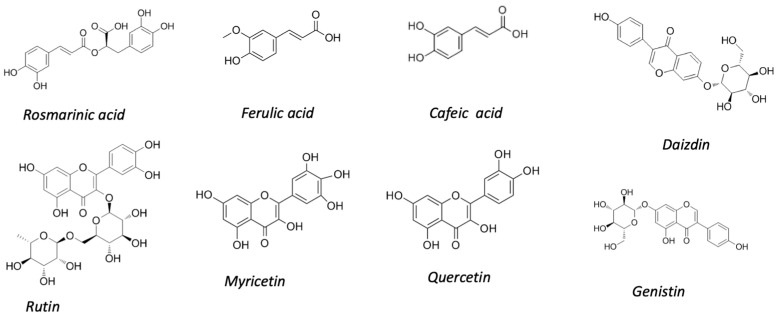
Chemical structure of the main polyphenolic compounds identified in chia seeds.

**Figure 3 plants-09-01359-f003:**
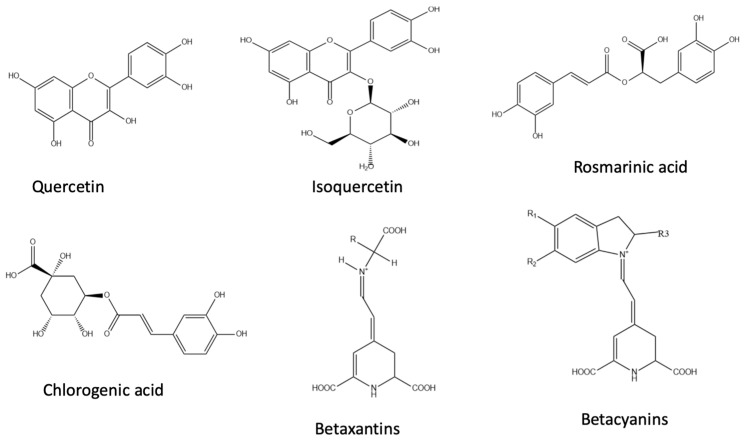
Chemical structure of the main antioxidant compounds identified in quinoa seeds.

**Table 1 plants-09-01359-t001:** Total phenolic content and antioxidant activity of chia, quinoa, and their coproducts.

	Origin	Total Phenol Content (TPC)	Antioxidant Activity		Extraction	References
		(mg GAE/g)	DPPH (mg TE/g)	FRAP (mg TE/g)	Protocol	
QUINOA						
**seeds** (white, red, and black)	Bolivia	5.0–6.6	3.4	2.7	double extraction (methanol/water and acetone/water)	Ballester-Sánchez et al. [12]
**seeds** (colored)	Peru	1.2–3.4	-	-	double extraction (methanol/water and acetone/water)	Abderrahim et al. [26]
**seeds** (red and yellow)	-	1.4–2.1	-	6–9	double extraction (methanol/water and acetone/water)	Brend et al. [61]
**seeds**	Bolivia	3.8	9.7	-	double extraction (methanol/water and acetone/water)	Pasko et al. [62]
**seeds** (white and colored)	Bolivia and Perú	3.9–4.2	1.9–5.0	2.4–4.6	double extraction (methanol/water and acetone/water)	Pellegrini et al. [4]
**seeds** (red)	Bolivia	4.9	6.0	-	double extraction (methanol/water and acetone/water)	Ballester Sánchez et al. [10]
**seeds** (white, red, and black)	Andean Region and Canada	2.0–5.2	1.3–2.8	2.0–8.8	one extraction (acetone/water)	Tang et al. [18]
**seeds**	Peru	0.8–3.4	1.6–2.2	-	double extraction (ethanol/water and methanol/water)	Valencia et al. [63]
**seeds** (white, red, and black)	Bolivia	5.1–6.1	1.8–3.5	-	double extraction (methanol/water)	Liu et al. [49]
**seeds** (yellow)	Poland, Denmark, Chile and Argentine	7.1–10.6	-	-	triple extraction (methanol/water, methanol/water and acetone/water)	Sobota et al. [64]
**seeds**	Peru	1.4–1.9	0.1–2.4	-	one extraction (ethanol/water)	Repo-Carrasco-Valencia et al. [65] Repo-Carrasco-Valencia and Celada [66]
**seeds** (white, red, and black)	Peru	0.6–1.0	0.5–1.5	-	one extraction (ethanol/water)	Díaz-Valencia et al. [67]
**seeds**	Iran	0.2–0.4	-	-	one extraction (ethanol/water)	Farajzadeh et al. [68]
**seeds**	Ecuador and Peru	7.7–8.6	0.7–1.7	0.7–2.2	triple extraction (methanol/water)	Dini et al. [48]
**flour** (white)	Mexico	1.8–3.2	-	-	one extraction (methanol/water)	Vazquez-Luna et al. [69]
**flour** (white)	Egypt	0.2	-	-	one extraction (methanol/water)	Sohaimy et al. [70]
**flour** (white, red, and black)	Peru	0.6–1.0	-	-	one extraction (ethanol/water)	Aguilar et al. [71]
**malted flour** (white, red, and black)	Peru	0.9–1.5	-	-	one extraction (ethanol/water)	Aguilar et al. [71]
**fiber-rich fraction by wet-milling** (red)	Bolivia	3.8	3.6	3.6	double extraction (methanol/water and acetone/water)	Ballester Sánchez et al. [10]
**fiber-rich fraction by dry-milling** (red)	Bolivia	6.6	5.2	6.9	double extraction (methanol/water and acetone/water)	Ballester Sánchez et al. [10]
**CHIA**						
**seeds**	Bolivia and Peru	4.1	5.6	70.1	double extraction (methanol/water and acetone/water)	Fernández-López et al. [22]
**seeds**	Australia	2.4	-	-	ethanol/water	Ding et al. [21]
**seeds**	Bolivia and Peru	3.9	5.4	71.8	double extraction (methanol/water and acetone/water)	Pellegrini et al. [3]
**seeds**	Bolivia and Chile	1.2	-	18.5	double extraction (methanol/water)	Oliveira-Alves et al. [60]
**seeds**	Chile	0.94	109.2	-	ethanol	Marinelli et al. [47]
**seeds**	Mexico	0.88–0.92	-	-	ethanol	Reyes-Caudillo et al. [23]
**seeds**	Italian	1.8	-	-	acetonitrile/acetic acid solution	Caruso et al. [72]
**seeds**	Mexico	0.5–0.7	-	-	ethanol	Porras-Loaiza et al. [73]
**seeds**	Mexico	1.64	-	-	triple (methanol/water)	Martínez-Cruz and Paredes-López [25]
**seeds**	Argentine	3.4	49.37		methanol/water	Tuncil and Celik [74]
**seeds**	Italy	1.5	1.6	-	ethanol/water	Antonini et al. [75]
**seeds**	Brazil	10.1–60.9	2.5–95.1	5.1–247.6	water/ethanol/acetone (alone and different mixtures)	Alcântara et al. [55]
**flour**	Brazil	4.8	-	-	acetonitrile/acetic acid solution	Dick et al. [76]
**flour**	Mexico	7.9	-	-	acetonitrile/acetic acid solution	Dick et al. [76]
**oil**	Bolivia and Chile	0.02	-	0.2	methanol/water	Oliveira-Alves et al. [60]
**oil**	Japan	4.9	6.1 IC_50_ (mg/L)	-	methanol	Xuan et al. [77]
**partially-deioled flour**	Bolivia and Chile	1.1	-	17.2	double extraction (methanol/water)	Oliveira-Alves et al. [60]
**partially-deioled flour**	Argentine	2.2	-	-	double extraction (acetone/water)	Aranibar et al. [13]
**partially-deioled flour**	Bolivia and Peru	5.0	7.0	81.0	double extraction (methanol/water and acetone/water)	Fernández-López et al. [22]
**partially-deioled flour**	Bolivia and Peru	4.9	7.2	80.9	double extraction (methanol/water and acetone/water)	Pellegrini et al. [3]

**Table 2 plants-09-01359-t002:** Chia and quinoa-based meat products and their effect on the lipid oxidation.

Meat Product	Chia/Quinoa	Effect on Lipid Oxidation (TBARs Value)	References
Frankfurter	chia seeds	35% reduction after 21 days refrigerated storage	Fernández-López et al. [9]
	chia flour	35% reduction after 21 days refrigerated storage	Fernández-López et al. [9]
	chia flour	300% increase (values < 0.2 mg MA/kg)	Pintado et al. [101]
	chia flour	not evaluated	Herrero et al. [102]
	deioled chia flour	35% reduction after 21 days refrigerated storage	Fernández-López et al. [9]
	chia oil (O/W emulsion)	250% increase (values < 0.2 mg MA/kg)	Pintado et al. [101]
	chia oil (O/W emulsion)	not evaluated	Herrero et al. [102]
	chia oil (emulsion gel)	275% increase (values < 0.2 mg MA/kg)	Pintado et al. [101]
	chia oil (emulsion gel)	not evaluated	Herrero et al. [102]
Ham-like product	chia seeds	30% reduction	Ding et al. [21]
Burgers	chia seeds	up to 68% reduction	de Oliveira-Paula et al. [103]
	chia seeds	50% reduction	Antonini et al. [75]
	deffated chia flour	increase TBARs	Souza et al. [96]
	chia oil	25% increase in raw burgers, 11% after cooking	Heck et al. [104]
	chia oil (microencapsulated)	50% increase in raw burgers, 70% after cooked	Heck et al. [104]
	chia oil (emulsion gel)	30% increase	Lucas-González et al. [105]
	chia oil (hydrogel emulsion)	up to 350% increase (at 100% substitution level)	Heck et al. [106]
	chia oil (O/W emulsion)	not evaluated	De Carvalho et al. [107]
	quinoa flour	not evaluated	Shokry [108]
	quinoa flour	20% reduction at 90 d frozen storage	Özer and Seçen [109]
	quinoa flour	not evaluated	Baioumy et al. [110]
Meatballs	quinoa flour	not evaluated	Bagdatli [111]
Fresh sausages	chia emulsion gel	not evaluated	Pintado et al. [112]
Cooked sausages	chia oil (O/W emulsion)	28% increase	De Carvalho et al. [113]
Nuggets	chia flour	not evaluated	Barros et al. [114]
	quinoa flour	not evaluated	Verma et al. [115]
Bolognas	chia flour	not evaluated	Pires et al. [116]
	chia emulsion gel	without variation	de Souza-Paglarini et al. [117]
	quinoa grains	50% reduction	Fernández-López et al. [98,99]
	quinoa flour	not evaluated	Vargas-Zambrano et al. [97]
	quinoa (fiber-rich fraction)	15% reduction	Fernández-López et al. [98,99]
Pâté	quinoa flour	20% reduction	Pellegrini et al. [118]
Dry-cured sausages	quinoa grains (boiled)	decrease hexanal content	Fernández-Díaz et al. [119]

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
