# Peer review of "Chia, Quinoa, and Their Coproducts as Potential Antioxidants for the Meat Industry"

_plants, 2020, doi:10.3390/plants9101359_

Round 1
Reviewer 1 Report
This is a good piece of work.
However, there are some minor, but highly important aspects, that authors need to consider before taking a decision regarding its full acceptation.
Abstract. It is a very descriptive part of this work; good description by the way. However, it contains only qualitatitve aspects and cero quantitative outstanding results.
Authors need to describe some quantitative and remarkable components of these two extremely important food crops. Otherwise, message to potential readers would be very limited.
Conclusions. Based on the whole components of this extensive work, and also based on the high number of references used for this study, authors need to indicate what type of research is needed for the future; what type of new studies are necessary to upgrade the scientific and technological potential of these so important Latin American food crops.
In brief, I do suggest its acceptation after the cited comments receive an adequate attention.
Ps. Authors are known in the scientific literature. In any case, it calls my attention that no author from Latin America is contributing to this work, when both crops have been extensively studied by scientists from the cited subcontinent. It is not exactly a requirement; however, the absence of any scientist from the cited region is noticeable.
Author Response
Reviewer 1
We thank all your comments and suggestions that allow us to clarify the message of our paper.
The paper has been carefully revised and language and grammatical errors have been corrected.
I am going to answer all your comments point by point. Your comments are in blue and my answers in black color.
This is a good piece of work.
Thanks a lot for your positive valuation about our work
However, there are some minor, but highly important aspects, those authors need to consider before making a decision regarding its full acceptance.
We thank your comments to improve the message of our paper and we will include your suggestions to enforce your decision regarding its acceptation
Abstract. It is a very descriptive part of this work; good description by the way. However, it contains only qualitative aspects and cero quantitative outstanding results.
Instructions for authors in this journal require that the abstract should be a total of about 200 words maximum. In my opinion, this qualitative information gives a general and complete vision which we consider more important than sacrificing some of this information to include quantitative data that have already been included throughout the paper.
Authors need to describe some quantitative and remarkable components of these two extremely important food crops. Otherwise, the message to potential readers would be very limited.
This information has been included in L54-61
L54-61 Nutritionally, quinoa stands out for its fiber content (5-11%), starch (52-74% being rich in amylopectin), proteins (13-17%) with presence of essential amino acids (methionine and lysine but also a high concentration of tryptophan, usually the second limiting amino acid in cereals) and fatty acids (2-9% oil with 90% unsaturated fatty acids of which 50-56% correspond to linoleic acid omega-6, 22-25% to oleic acid omega-9 and 5-7% to linolenic acid omega-3), absence of gluten, and adequate levels of key micronutrients, including minerals (calcium, magnesium, iron, potassium, phosphorus, manganese, zinc, copper and sodium) and vitamins (vitamin B complex, E and C) [8, 15-17]. Quinoa also contains bioactive constituents, such as carotenoids, tocopherols, polyphenols, and betalains, that are responsible for protecting the plant against adverse climatic conditions and are highly correlated with its antioxidant activity [4, 18, 19]. Regarding chia, it stands out also for its fiber content (18-30% being soluble dietary fiber 7-15%), proteins of high biological value (15-25%) and it is considered a natural source of omega-3 (linolenic acid up to 68% of fatty acid content) and omega-6 fatty acids (linoleic acid 19%) [20-22]. It also contains bioactive compounds such as tocopherols, polyphenols, vitamins (mainly vitamin B1, B2 and niacin), and carotenoids [23-25].
Conclusions. Based on the whole components of this extensive work, and also based on the high number of references used for this study, authors need to indicate what type of research is needed for the future; what type of new studies are necessary to upgrade the scientific and technological potential of these so important Latin American food crops.
Now we have included a paragraph at the end of conclusions to describe future views in this field.
“In addition to optimize the formulation to avoid technological problems and to ensure a positive sensorial evaluation of the final meat product, it could be interesting to assess the effect of digestion process on the stability and bioaccessibility of these bioactive compounds and their antioxidant properties using in vitro digestion studies. “
In brief, I do suggest its acceptance after the cited comments receive adequate attention.
I hope that now the paper will be adequate for publishing in this journal
Ps. Authors are known in the scientific literature. In any case, it calls my attention that no author from Latin America is contributing to this work, when both crops have been extensively studied by scientists from the cited subcontinent. It is not exactly a requirement; however, the absence of any scientist from the cited region is noticeable.
As you can see in the reference section a lot of works about Argentine, Peruvian, Bolivian, Mexican, and other Latin American chia and quinoa seeds have been consulted for this work. My research group is working with these seeds from some years ago and of course that we have been in touch with some scientists from Latin America, experts in this field. The work for this revision paper has been made only by my research group.
Reviewer 2 Report
Comments to the authors:
The present manuscript is a ver renewed review about two of the most popular food ingredients nowadays. This version is very well explained and distributed, and it complies with the criteria of the journal, it may have relevant scientific soundness, specially for those working in the development of novel meat formulations and meat products.
Nonetheless, it is as consideration for this reviewer that the authors must include some observations (specially concept definition/application of antioxidant activity) in order to improve this significant work.
__________
- It is missing the natural role of antioxidants in their original matrix. Meaning, i.e. most of the phenolic compounds in chia are known to act as a protection for the seed oil, which is highly oxidizable. Thus, depending of their composition, they will be wasted by protecting omega fatty acids from peroxidation before having a relevant biological activity in consumers (when using the whole seed or oil, not happening when using phenolic fraction).
- Figure 3. Chemical structure sizes must be homogenized.
- L323: More than antioxidant properties, this whole section is about oxidation stability of meat products containing chia or quinoa. Please consider.
- L374-385: Given the advances in food science and technology, this part of the manuscript can give to the present work an additional relevance. It is one of the most important points in discussion for new research. I believe it deserves to be included as a subsection (i.e. named <<formulation strategies in meat and meat products>>, or whatever the authors considered appropriate), and extend a bit explaining the different technologies applied to the incorporation of these ingredients into final products.
- In line with previous observation, a small clarification about the concept of functional food, and the difference with enriched products should be included in this subsection. When talking about inclusion of an ingredient into a preparation it is not the same if we aim to get a biological activity in consumers or jus adding new ingredients to gain stability/protection/higher conservation/etc.
- In general, even if the whole manuscript is well stablished, the use of <<antioxidant activity>> is not completely correct, the aim if the present work is to review the oxidation satiability that chia/quinoa and their coproducts give to meat and meat products. But is never mentioned the antioxidant activity (which is the antioxidant effect of the ingredient/product/coproduct consumption). I recommend the authors to change this concept.
Author Response
We thank all your comments and suggestions that allow us to clarify the message of our paper.
The paper has been carefully revised and language and grammatical errors have been corrected.
I am going to answer all your comments point by point. Your comments are in blue and my answers in black color.
Comments to the authors:
The present manuscript is a very renewed review about two of the most popular food ingredients nowadays. This version is very well explained and distributed, and it complies with the criteria of the journal, it may have relevant scientific soundness, specially for those working in the development of novel meat formulations and meat products.
Thanks for your positive valuation about our work
Nonetheless, it is as consideration for this reviewer that the authors must include some observations (specially concept definition/application of antioxidant activity) in order to improve this significant work.
Of course, we thank you the corrections to help us to improve the quality of our work and also to clarify some misleading
- It is missing the natural role of antioxidants in their original matrix. Meaning, i.e. most of the phenolic compounds in chia are known to act as a protection for the seed oil, which is highly oxidizable. Thus, depending of their composition, they will be wasted by protecting omega fatty acids from peroxidation before having a relevant biological activity in consumers (when using the whole seed or oil, not happening when using phenolic fraction).
Accepting that the first objective of these compounds in their original matrix is to protect some compounds highly oxidizable (in the case of chia oil as in the case of olive oil) it is also true and widely accepted that their consumption can have beneficial effect on the health mainly attributed to these bioactive compounds.
- Figure 3. Chemical structure sizes must be homogenized.
- This figure has been homogeneized
- L323: More than antioxidant properties, this whole section is about oxidation stability of meat products containing chia or quinoa. Please consider.
You are right. It has been changed and now it is “Oxidation stability of meat products containing chia, quinoa, or their coproducts”
- L374-385: Given the advances in food science and technology, this part of the manuscript can give to the present work an additional relevance. It is one of the most important points in discussion for new research. I believe it deserves to be included as a subsection (i.e. named <<formulation strategies in meat and meat products>>, or whatever the authors considered appropriate), and extend a bit explaining the different technologies applied to the incorporation of these ingredients into final products.
I agree with you that this part is very interesting about the strategies for the incorporation of vegetable oils in meat products but it is not the aim of this review and it is not specific for the chia oil but for others vegetable oils with heathier lipid profile than animal fats to be used as fat substitutes. So, taking into account the extension of this review we have reported only some strategies applied in the formulation of meat products with chia oils but there are others that have not been commented because no references about their application in meat products with chia oil added have been found.
- In line with previous observation, a small clarification about the concept of functional food, and the difference with enriched products should be included in this subsection. When talking about inclusion of an ingredient into a preparation it is not the same if we aim to get a biological activity in consumers or jus adding new ingredients to gain stability/protection/higher conservation/etc.
We have revised the manuscript to found where the confusion between these two concepts (functional food vs enriched food) could have been induced.
In this paragraph we can check that it has been correctly used.
L388-391 “Quinoa has been mainly applied in meat products (burgers, meatballs, nuggets, bolognas, pâté, dry-cured sausages, etc.) as techno-functional ingredient (due to its high protein and carbohydrate content) or as fat replacer (due to its behavior as a fat-like raw material). In the first case (techno-functional ingredient……”
It is true that in this other paragraph it could be misleading so it has been changed
L342-343 “This addition was taken off with two main objectives: to partially replace animal fat or to improve their nutritional and functional quality” in this case the word functional has been changed by techno-functional
- In general, even if the whole manuscript is well stablished, the use of <<antioxidant activity>> is not completely correct, the aim if the present work is to review the oxidation satiability that chia/quinoa and their coproducts give to meat and meat products. But is never mentioned the antioxidant activity (which is the antioxidant effect of the ingredient/product/coproduct consumption). I recommend the authors to change this concept.
I can understand that in some specific points these two concepts have been misleading ( and now they have been corrected as you suggested) but we consider that it has been well defined in the description of the objective,
L90-L94 “Considering that, in this review, we summarize and discuss recent advances about the composition and identification of bioactive compounds with antioxidant properties in chia and quinoa edible grains and their coproducts. It is also aimed at exploring the effect of the addition of chia, quinoa or their coproducts on the development of the lipid oxidation in chia and quinoa related meat products due to the antioxidant properties attributed to these bioactive compounds”.
An also in this case L64-65 “On the other hand, the addition of chia or/and quinoa in foods enables that their antioxidant compounds could prevent food deterioration caused by lipid oxidation”
In this other, a small change has been included
L368-369 “which is showing the effect of inhibiting lipid oxidation, suggesting their use as natural antioxidant” has been changed by “which is showing the effect of inhibiting lipid oxidation, suggesting their use as natural antioxidant in meat products”
Reviewer 3 Report
The review “Chia, quinoa, and their coproducts as potential antioxidants for the meat industry” provides very valuable information on the antioxidant compounds of chia and quinoa coproducts and on the strategies used to add them to meat products highlighting their effect on the lipid oxidation control.
The paper is well presented and written in good language and style. In conclusion, I would recommend the paper for publication in Plants in present form.
Author Response
We are very happy that our work has been so positively scored for you. Thanks a lot.
Round 2
Reviewer 2 Report
The authors have improved their manuscript by following referees suggestions and keeping the line of their work by refuting the reviewers observations.
It is for consideration of this referee that thae manuscript can be accepted in this present form.